# Beyond individual choice: Exploring pathways linking couple dynamics with Unintended Pregnancy and Birth in India

**Monika Lakhotia**[iD]*, **Bhaswati Das**[iD]

Centre for the Study of Regional Development, Jawaharlal Nehru University, Munirka, New Delhi, India

* monikalakhotia646@gmail.com

## Abstract

Couple power dynamics are known to influence sexual agency and reproductive health, yet this negotiation process within marital relationships remains empirically underexamined in India's public health discourse. This study investigates Unintended Pregnancy and Birth (UPB) as an outcome associated with relational systems involving husbands' sexual attitudes (normative context), wives' health autonomy (negotiated agency), and Intimate Partner Violence (IPV; structural constraint). Using a sample of 19,896 couples from NFHS-5 (2019−21), we conducted bivariate and multivariate logistic regressions to identify patterns of UPB. Subsequently, a Generalized Structural Equation Model (GSEM) was used to test the mediating role of wives' health autonomy on the pathway between husbands' attitudes and UPB. To explore the moderating role of IPV between wives' health autonomy and UPB, we used an interaction term. Findings show that overall, 8.30% of pregnancies and births are unintended. While wives' health autonomy significantly reduces the risk of UPB, IPV act as the persistent structural constraint that elevates UPB. Furthermore, husbands' regressive sexual attitudes, reflecting a normative climate of sexual entitlement, indirectly increased the likelihood of UPB by significantly restricting wives' health autonomy. These findings highlight how intimate power asymmetries and normative constraints shape reproductive experiences. Addressing UPB, which reflects compromised autonomy and gender control, requires gender-transformative approaches that go beyond individual interventions to challenge inequitable norms, engage both partners and strengthen agency within intimate relationships to support reproductive well-being.

## Introduction

The International Conference on Population and Development (ICPD) in 1994 marked a paradigmatic shift in addressing sexual and reproductive health by situating it within a human rights and choice-based framework. Reproductive health was

**Data availability statement:** The data underlying this study can be accessed from https://www.dhsprogram.com/data/available-datasets.cfm.

**Funding:** The author(s) received no specific funding for this work.

**Competing interests:** The authors have declared that no competing interests exist.

defined as "a state of complete physical, mental and social well-being and not merely the absence of disease or infirmity" in matters relating to the reproductive system [1]. Moving beyond an individualistic focus on women alone, the ICPD emphasized the inclusion of men not only as clients but also as agents of positive change in sexual and reproductive health [2]. This rights-based framing was further reinforced by the Guttmacher–Lancet Commission, which highlighted sexual and reproductive health as central to overall well-being and sustainable development [3].

Since then, global scholarship has increasingly emphasized gender power dynamics as fundamental to reproductive health outcomes. Gender norms embedded within social structures shape power relations across levels, from intra-household negotiations to broader development contexts. Recognizing power imbalances through an intersectionality lens is essential for understanding how negotiation within couples' shapes health and well-being [4,5] and for advancing gender equality goals under the Sustainable Development Agenda [6]. Recent work distinguishing between a wife "taking power" and being "given power" further demonstrates that agentic forms of power are particularly beneficial for reproductive health [7].

Reproductive health outcomes emerge from multidimensional and relational processes rather than linear individual choices [8,9]. However, research and policy interventions have often remained women-centric, prioritizing access to services while neglecting men's engagement and relational contexts, thereby limiting the long-term effectiveness of family planning programs [10]. Gender-hegemonic frameworks may further exacerbate women's vulnerabilities by reinforcing unequal power relations [11]. Empirical evidence underscores the importance of men's involvement, with studies showing improved maternal and reproductive outcomes when male partners are meaningfully engaged [12]. Gender inequity operates as a critical social determinant of health, linking gender relations and health outcomes in deeply intertwined ways [13–15]. Men's sexual and attitudinal norms, those endorsing control or violence, and women's capacity to make informed reproductive choices represent key dimensions of couple-level interaction influencing reproductive health [16]. Empowerment, defined as the expansion of the ability to make strategic life choices in contexts where such ability was previously denied [17], can facilitate decisions regarding the timing and spacing of births and improve family well-being [18]. Yet empowerment is situational and may not always act as a protective factor; women endorsing traditional gender norms may adopt submissive roles that heighten reproductive risks, including unintended pregnancy [16,19].

In India, these dynamics are particularly salient. Male-dominated decision-making, social constraints, and fear of violence often prevent women from translating reproductive preferences into effective contraceptive use [20]. Women's capacity to negotiate reproductive choices and resist intimate partner violence is closely tied to men's attitudes [21]. Intra-household power relations thus play a pivotal role in shaping reproductive decisions [22], while broader social relations influence individuals' ability to act within these negotiations [23]. In such contexts, individual empowerment alone may not be sufficient and relational empowerment embedded within wider social and political structures becomes critical [17]. This helps explain the persistence of gender-based

violence globally, with one in three women experiencing physical or sexual violence in their lifetime [24]. Empirical evidence demonstrates that sexual and intimate partner violence significantly increases the risk of unintended pregnancy [25], while women's autonomy may partially mediate this relationship [26]. Together, these findings underscore that relational power is context-dependent and central to understanding reproductive risks such as unintended pregnancy, unwanted birth, and other adverse outcomes. India, despite being the first country to launch a national family planning programme in 1952 and achieving substantial fertility decline, continues to face persistent reproductive risks. More than one in ten women experience unintended pregnancy, alongside consistently low rankings in global gender equality indices [27]. While qualitative studies document the influence of gender norms and power asymmetries on reproductive decision-making, empirical research often relies on unitary frameworks that obscure relational processes, limiting policy effectiveness. Against this backdrop, the present study empirically examines how Unintended Pregnancy and Birth (UPB) is shaped by couple-level relational dynamics, focusing on husbands' sexual attitudes, wives' health autonomy, and intimate partner violence.

## Theoretical framework

This study examines the outcome 'Unintended Pregnancy and Birth' (UPB) by focusing on relational characteristics of couples. By Synthesizing Connell's theory of Gender and Power [28] and its extension in public health by Wingood & DiClemente [29] Kabeer's empowerment framework [17], and Heise's ecological model of violence [30], we frame UPB as potentially associated with three interrelated dimensions: (i) normative context, (ii) process of negotiated agency, and (iii) structural constraints (Fig 1). Building on these theories, we move from broad social theory to the specific relational pathways associated with UPB.

**Normative context: Husband's sexual attitude.** We begin with the premise that every couple exists within a specific normative context. Following Connell and Wingood & DiClemente, we conceptualise norms surrounding sexual entitlement and refusal, often discussed under the domain of cathexis, as central to gender relations and outcomes within intimate partnerships. In Kabeer's terminology, these norms function as a resource (or a lack thereof) that shapes the environment for choice. In this study, the normative context is operationalised through the presence of inequitable sexual attitudes held by the husband. Theoretically, these attitudes represent 'rules of intimacy' that define whether a wife's preferences are validated or marginalized. We test whether the presence of such attitudes is associated with a higher likelihood of UPB.

**The negotiation process: Wives' health autonomy.** The second dimension of our framework is agency, which Kabeer defines as the process by which individuals define and act on their goals [17]. Within couple dynamics, agency is a negotiated capacity between partners. We measure this negotiation process through the presence of wives' health autonomy. Theoretically, we also position this agency as a potential mediator: the husband's attitudes (the normative context) may shape the wife's capacity to exercise autonomy (the negotiation process) and thus influence the outcome UPB.

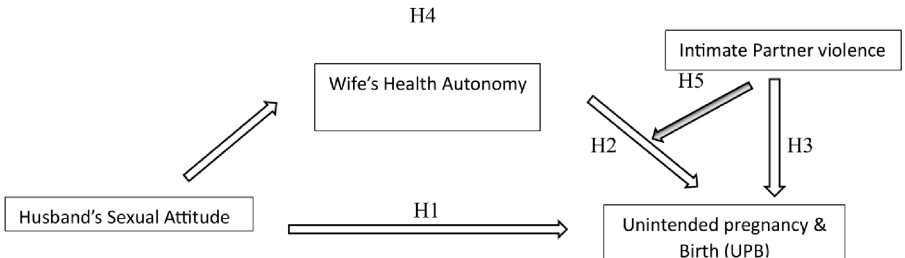

**Fig 1. Pathways linking couple dynamics and UPB.**

**The structural constraint: Intimate partner violence.** We acknowledge that agency does not function in a vacuum. Following the Ecological Model of Violence and Reproductive Health [30] we incorporate intimate partner violence (IPV) as a structural constraint within the relationship. IPV may shape the environment in which negotiation takes place by restricting communication, increasing fear, or limiting wives' ability to act on their preferences. We therefore consider IPV both as a factor associated with UPB and as a condition that may alter how wives' autonomy relates to UPB. With this background, we hypothesise that-

- H1 (Normative Association): The presence of inequitable sexual attitudes among husbands is associated with a higher likelihood of UPB.

- H2 (Agency Association): wives' health autonomy is associated with a lower likelihood of UPB.

- H3 (Constraint Association): Experience of intimate partner violence by wives is associated with a higher likelihood of UPB.

- H4 (Mediation Pathway): Wives' health autonomy is hypothesised to mediate the association between husbands' attitudes and UPB, representing the path of negotiated power.

- H5 (Moderation Pathway): The association between wives' health autonomy and UPB is hypothesized to be moderated by IPV.

## Materials and methods

### Data source

We used the information from the National Family Health Survey 5 (NFHS 5; 2019−21) for the present study. It is a large-scale, nationally representative, cross-sectional survey conducted across India by the Ministry of Health and Family Welfare, Government of India. NFHS-5 employed a two-stage stratified cluster sampling design and covered all 29 states and 7 Union Territories, ensuring both national and sub-national representativeness. In total, the survey collected information from 636,699 households, including 724,115 women (aged 15–49 years) and 101,839 men (aged 15–54 years). For this study, the analytical sample was restricted to currently married couples in which the wife had experienced at least one live birth or was pregnant at the time of the survey. After excluding cases with missing information, the analytical sample consisted of 19,896 couples. When incorporating measures of intimate partner violence (IPV), which were available only for a sub-sample, the effective sample size was reduced to 16,168 couples. All analyses were conducted using the appropriate sample weights to preserve national representativeness and adjust for the survey design. As the study used secondary data available in the public domain upon request, no additional ethical approval was required.

### Study variables

**Dependent variable.** The outcome variable is Unintended Pregnancy and Birth (UPB), coded as binary (No or Yes) based on wives' responses. In NFHS, those who were pregnant at the time of survey were asked whether their pregnancy was wanted at that time, wanted later, or not wanted at all. Women who had a live birth in the five years preceding the survey were similarly asked whether their last birth was wanted at that time, wanted later, or not wanted at all. Pregnancies or births reported as "wanted later" or "not wanted at all" were classified together as unintended pregnancy and birth (UPB).

To ensure the robustness of the UPB measure, we conducted separate bivariate analyses for its two components, current unintended pregnancy and last unintended birth, using the Rao–Scott chi-square test to examine their associations with sociodemographic, economic, and relational variables (S1 Table).

**Explanatory variables.** Guided by our theoretical framework, we selected three couple-level relational characteristics as three key explanatory variables: wives' health autonomy, husbands' sexual attitude, and experience of intimate partner violence (IPV).

**'Wives' health autonomy'** was measured using information collected from wives on who usually makes decisions regarding their health care. Responses were categorised as "Yes" if the wife reported that the decision was made by herself alone or jointly with her husband, and "No" if the decision was made by the husband alone, someone else or reported other.

An index was created to measure **'Husband's sexual attitude'**. Husbands were asked if their wife refuses to have sex, are they justified to (i) get angry, (ii) refuse financial support, (iii) force sex, or (iv) have sex with another woman. Husbands who responded "Yes" or "don't know, depends" to any of these items were coded as having a regressive attitude, while those responding "No" to all items were coded as progressive. Husbands were additionally asked whether a wife is justified in refusing sex when (i) they have a sexually transmitted infection, (ii) have sexual relations with other women, or (iii) the wife is tired or not in the mood. For these items, responses of "No" or "Don't know" to any item were coded as regressive, while "Yes" responses were coded as progressive. Husbands were classified as having a regressive sexual attitude if they gave a regressive response to any one or more of the first four items and any one or more of the last three items; otherwise, they were coded as progressive. Inter-item correlations were assessed to ensure conceptual coherence and suitability for index construction.

**'Intimate Partner Violence' (IPV)** was measured using the NFHS-5 domestic violence module, which includes items on physical, sexual, and emotional violence. Physical violence items assessed whether women had ever been pushed, slapped, punched, kicked, strangled, burnt, or threatened with a weapon by their husband or partner. Sexual violence items included whether women had ever been physically forced into unwanted sex, forced to perform sexual acts, or physically forced to unwanted sexual acts by their husband or partner. Emotional violence items included whether women had ever been humiliated, insulted, or threatened by their husband or partner. Wives' reporting experience of any form of violence, alone or in combination, was classified as having experienced IPV.

Additionally, a range of **sociodemographic and economic** variables was included [31,32]. These are **age, educational attainment, occupation, exposure to contraceptives in mass media, religion,** and **social group** for **both partners** (husband and wife), whereas the cited studies typically considered only women's characteristics. **Household wealth quintile, place of residence** (rural or urban**), years of marriage,** and **number of living children** were also included to capture broader socioeconomic and reproductive context.

**Data analysis.** We first conducted bivariate analyses to examine the distribution of UPB across selected sociodemographic and economic characteristics, as well as across the three key explanatory variables related to couple dynamics, namely husbands' sexual attitude, wives' health autonomy, and experience of intimate partner violence (IPV). Associations were assessed using the Rao–Scott chi-square test. Four logistic regression models were estimated for the outcome UPB. The first three models assessed the association between UPB and each of the key explanatory variables, namely husbands' sexual attitude, wives' health autonomy, and wives' experience of IPV. We then estimated a full multivariable model that included all sociodemographic and economic variables that were statistically significant at the 10% level ($p < 0.10$) in bivariate analysis, along with the three key explanatory variables. Further to examine the relational pathways associated with UPB, we used Generalised Structural Equation Modelling (GSEM). This approach was used to assess whether husbands' sexual attitude was directly or indirectly associated with UPB through its relationship with wives' health autonomy. In this model, husbands' sexual attitude was specified as the independent variable, wives' health autonomy as the mediator, and all other variables were controlled. Following Hayes [33], full mediation was considered when the indirect pathway through autonomy was statistically significant, and the direct association between husbands' sexual attitude and UPB was not; partial mediation was considered when both direct and indirect associations were statistically significant. Then we examined whether violence can moderate the association between wives' health

autonomy and UPB. This was assessed by estimating an interaction term between women's health autonomy and IPV within a logistic regression framework, controlling for all variables. The interaction effect was further probed by plotting predicted probabilities of unintended pregnancy across different combinations of wives' health autonomy and IPV.

## Results

### Sample characteristics and UPB patterns

Table 1 presents the socio-demographic, economic, and relational characteristics of couples included in our analysis, along with the distribution of unintended pregnancy and birth (UPB) across these characteristics. More than half of the couples in the analysis are in the 25–34-year age group. Around one-third of couples reside in rural areas, while the remaining live in urban areas. Regarding education, 18.46% of wives have no formal education, 11.70% have primary education, 52.90% have secondary education, and 16.94% have completed higher education. Among husbands, 11.79% have no formal education, 18.82% have completed higher education, and the majority (56.49%) have completed secondary education. The majority of wives in the analysis are Hindu (83.22%), followed by Muslim (12.02%), Christian (2.15%), and other religions (2.61%). A similar distribution is observed among husbands. By social group, Other Backward Classes constitute the largest share (45.38%), followed by Scheduled Castes (24.33%), Others (18.69%), and Scheduled Tribes (11.60%). In terms of occupation, 75.92% of wives are not working, whereas only 2.90% of husbands are not working. 78.71% of wives report having health autonomy in making their own health decisions, while about 21% depended on their husbands or others. Additionally, 44.36% of husbands' responses are regressive on sexual attitudes. Nearly one in three wives has experienced any form of IPV.

Overall, 8.30% wives have experienced UPB. The prevalence is higher among wives aged 35 and above or husbands aged 45 and above. UPB is more common among wives with primary education, those working in skilled or unskilled manual occupations, and those with lower exposure to mass media for contraceptive use. Similarly, higher UPB has been experienced by wives whose husbands have no formal education, were engaged in skilled or unskilled manual work, or lacked exposure to mass media on contraception. Couples in the poorest wealth quintile, those residing in rural areas, and those belonging to the Muslim religion have a higher prevalence of UPB. It is also more prevalent among couples married for more than 25 years (14.48%) and among those with three or more living children (14.80%). Wives without health autonomy have higher UPB (10.31%), as do those who have experienced IPV (11.02%). UPB is higher when husbands' responses are regressive on sexual attitude (8.74%).

### Associates of UPB among sampled couples

Table 2 presents the results of the logistic regression analyses. Models 1–3 report unadjusted associations between key explanatory variables and unintended pregnancy and birth (UPB). Model 1 indicates a significant association between wives' health autonomy and UPB. Wives

who have autonomy in making their own health decisions have significantly lower odds of UPB compared to those without such autonomy (OR = 0.75, p < 0.01). Model 2 shows that husbands' sexual attitude is not significantly associated with UPB. Model 3 indicates that wives who ever experienced any form of intimate partner violence have significantly higher odds of UPB compared to those who do not (OR = 1.47, p < 0.01).

Model 4 is the fully adjusted model, which includes socio-demographic and economic variables along with other relational characteristics to examine adjusted associations with UPB. Findings highlight that after adjustment, the odds for UPB for wives experiencing violence decreased from 1.47 to 1.24 times. The protective association of wives' health autonomy is slightly weakened; the odds ratio increased from 0.73 to 0.75. Regarding socio-demographic characteristics, wives aged 25–34 years are less likely to have UPB compared to those under 25 years (OR = 0.71, p < 0.01). Wives with primary, secondary, or higher education faced increased odds of UPB relative to those with no formal education (OR = 1.45,

**Table 1. Patterns of UPB by socio-demographic, economic, and relational characteristics of couples.**

| Characteristics | | % of UPB | Total number of couples (wives' currently pregnant or had a live birth) |
|---|---|---|---|
| | | 8.30% (weighted) | (Unweighted n = 19896) |
| **Wife Age group\*\*\*** | 15-24 | 7.75 | 5941 |
| | 25-34 | 7.95 | 11753 |
| | 35 −49 | 12.36 | 2202 |
| **Husband Age Group\*\*\*** | 15-24 | 7.59 | 1818 |
| | 25-34 | 7.96 | 11619 |
| | 35-44 | 8.83 | 5645 |
| | 45 −54 | 11.68 | 814 |
| **Wife's level of education\*\*\*** | no education | 10.59 | 3900 |
| | primary | 10.47 | 2458 |
| | secondary | 7.77 | 10648 |
| | higher | 5.97 | 2890 |
| **Husband's educational level\*\*\*** | no education | 11.30 | 2515 |
| | primary | 10.25 | 2510 |
| | secondary | 7.90 | 11417 |
| | higher | 6.30 | 3454 |
| **Wife's Occupation\*\*** | not working | 8.51 | 14421 |
| | salaried | 7.26 | 1356 |
| | agricultural | 7.20 | 2970 |
| | skilled & unskilled manual | 8.95 | 1149 |
| **Husband's Occupation** | not working | 7.79 | 638 |
| | salaried | 7.31 | 5023 |
| | agricultural | 8.65 | 7280 |
| | skilled & unskilled manual | 8.76 | 6955 |
| **Mass media exposure for contraceptives (wife)\*\*\*** | no | 9.69 | 7365 |
| | yes | 7.57 | 12531 |
| **Mass media exposure for contraceptives (Husband)** | no | 9.23 | 6586 |
| | yes | 7.91 | 13310 |
| **Household wealth quintile\*\*\*** | poorest | 11.33 | 4678 |
| | poorer | 8.71 | 4456 |
| | middle | 7.98 | 4002 |
| | richer | 7.21 | 3728 |
| | richest | 6.15 | 3032 |
| **Wife's Religion** | Hindu | 8.17 | 15015 |
| | Muslim | 9.38 | 2169 |
| | Christian | 8.12 | 1781 |
| | others | 7.68 | 931 |
| **Husband's Religion** | Hindu | 8.25 | 15017 |
| | Muslim | 9.07 | 2187 |
| | Christian | 7.22 | 1738 |
| | others | 7.26 | 954 |
| **Wife's Caste** | SC | 8.87 | 4100 |
| | ST | 7.46 | 4675 |
| | OBC | 8.19 | 7780 |
| | Others | 8.35 | 3341 |

*(Continued)*

**Table 1.** (Continued)

| Characteristics | | % of UPB | Total number of couples (wives' currently pregnant or had a live birth) |
|---|---|---|---|
| **Husband's Caste** | SC | 9.36 | 4130 |
| | ST | 7.00 | 4646 |
| | OBC | 8.14 | 7797 |
| | Others | 8.11 | 3323 |
| **Place of Residence\*\*\*** | rural | 8.81 | 15301 |
| | urban | 7.10 | 4595 |
| **Years of marriage\*\*\*** | upto5years | 6.25 | 5355 |
| | 6-10years | 7.65 | 7861 |
| | 11to15years | 8.71 | 4078 |
| | 16to25 years | 14.01 | 2328 |
| | >25years | 14.48 | 274 |
| **Number of living Children\*\*\*** | no child | 3.95 | 1163 |
| | 1 or 2 children | 6.27 | 13308 |
| | 3 or more children | 14.80 | 5425 |
| **Husband's Sexual Attitude** | Progressive | 7.96 | 10831 |
| | Regressive | 8.74 | 9065 |
| **Wife's health Autonomy\*\*\*** | no | 10.31 | 4019 |
| | yes | 7.76 | 15877 |
| **Violence\*\*\*** | no | 7.82 | 11602 |
| | yes | 11.02 | 4566 |

\*\*\* $p < 0.01$, \*\* $p < 0.05$, \* $p < 0.1$.

1.46, and 1.57, respectively; $p < 0.01$). Wives engaged in agricultural work have a reduced likelihood of UPB compared to non-working wives (OR = 0.69, $p < 0.01$). Household wealth exhibits a clear gradient, with couples in progressively higher quintiles showing lower odds of UPB than those in the poorest quintile. No of living children is strongly associated with UPB. Couples with one or two living children have almost twice the odds of UPB (OR = 1.89, $p < 0.01$), and those with three or more children have substantially higher odds (OR = 5.16, $p < 0.01$) compared to childless couples.

## Results of the mediation analysis

The results of assessing linkages between husbands' sexual attitudes, wives' health autonomy, and UPB have been presented in Table 3. The result indicates a statistically significant negative association between husbands' regressive sexual attitude and wives' health autonomy (Path a; β = −0.18, $p < 0.01$). Again, Wives' health autonomy is negatively associated with UPB (path b; β = −0.39, $p < 0.01$), indicating that having health autonomy is associated with a lower

likelihood of UPB. No direct statistically significant effect has been found of the husband's sexual attitude on UPB (path c′; β = 0.03). However, there is a significant indirect positive effect (β = 0.07, $p < 0.01$) of husbands' sexual attitudes on UPB through wives' health autonomy, indicating a pattern of full statistical mediation between husbands' sexual attitudes and UPB.

## Results of the moderation analysis

The predicted probabilities of UPB across levels of wives' health autonomy, stratified by experience of intimate partner violence, are presented in Fig 2. While women with health autonomy exhibit lower predicted probabilities of UPB in both

**Table 2. Factors associated with UPB among currently married couples.**

| Variables | | Model 1 Odds ratio (CI) | Model 2 Odds ratio (CI) | Model 3 Odds ratio (CI) | Model 4 Odds ratio (CI) | Model 5 Odds ratio (CI) |
|---|---|---|---|---|---|---|
| **Wife's health autonomy, No®** | Yes | | 0.73*** (0.62 - 0.86) | | | 0.75*** (0.62 - 0.90) |
| **Husband's Sexual Attitude, Progressive®** | Regressive | | | 1.11 (0.95 - 1.29) | | 1.12 (0.95 - 1.33) |
| **Violence, No®** | Yes | | | | 1.47*** (1.24 - 1.74) | 1.24** (1.04 - 1.47) |
| **Age group (Wife) <25years®** | 25–34 | | | | | 0.71*** (0.56 - 0.90) |
| | 35 - 49 | | | | | 0.93 (0.62 - 1.39) |
| **Age group (Husband) <25years®** | 25–34 | | | | | 0.94 (0.64 - 1.36) |
| | 35–44 | | | | | 0.80 (0.53 - 1.21) |
| | 45 - 54 | | | | | 0.70 (0.40 - 1.22) |
| **Education (Wife) Illiterate®** | Primary | | | | | 1.45*** (1.13 - 1.87) |
| | Secondary | | | | | 1.46*** (1.17 - 1.83) |
| | Higher | | | | | 1.57** (1.10 - 2.23) |
| **Education (Husband) Illiterate®** | Primary | | | | | 0.98 (0.75 - 1.27) |
| | Secondary | | | | | 0.89 (0.70 - 1.12) |
| | Higher | | | | | 0.90 (0.63 - 1.30) |
| **Working Status (Wife) Not working®** | salaried | | | | | 1.00 (0.72 - 1.40) |
| | agricultural | | | | | 0.69*** (0.55 - 0.87) |
| | skilled & unskilled manual | | | | | 1.03 (0.72 - 1.48) |
| **Mass media exposure for contraceptive (Wife), No®** | yes | | | | | 0.98 (0.82 - 1.17) |
| **Household Wealth quintile, Poor®** | Poorer | | | | | 0.83* (0.68 - 1.02) |
| | Middle | | | | | 0.79* (0.61 - 1.01) |
| | Richer | | | | | 0.74** (0.57 - 0.95) |
| | Richest | | | | | 0.62*** (0.44 - 0.89) |
| **Place of residence, Rural®** | Urban | | | | | 1.01 (0.80 - 1.26) |

*(Continued)*

| | | Model 1 | Model 2 | Model 3 | Model 4 | Model 5 |
|---|---|---|---|---|---|---|
| **Years Since Marriage, <6 years®** | 6-10years | | | | | 1.04 (0.80 - 1.35) |
| | 11to15years | | | | | 0.88 (0.64 - 1.22) |
| | 16to25 years | | | | | 1.34 (0.91 - 1.97) |
| | >25years | | | | | 1.37 (0.77 - 2.43) |
| **Number of living children, No child®** | 1 or 2 children | | | | | 1.89** (1.12 - 3.20) |
| | 3 or more children | | | | | 5.16*** (2.94 - 9.06) |
| **Constant** | | 0.08*** (0.08 - 0.09) | 0.09*** (0.08 - 0.10) | 0.11*** (0.10 - 0.13) | 0.08*** (0.07 - 0.09) | 0.05*** (0.03-0.09) |
| **Observations** | | 19,896 | 19,896 | 19,896 | 16,168 | 16,168 |

˚Reference category.

*** p<0.01, ** p<0.05, * p<0.1.

**Table 3. Pathways linking husbands' sexual attitudes, wives' health autonomy, and UPB.**

| Path | Estimate (β) |
|---|---|
| Path a: Husband's sexual attitude → Wife's health autonomy | −0.18*** |
| Path b: Wife's health autonomy → UPB | −0.39*** |
| Path: c' Husband's sexual attitude → UPB (direct effect) | 0.03 |
| Indirect effect (a*b) | 0.07*** |

*** p<0.01, ** p<0.05, * p<0.1.

groups, women who experienced violence consistently show higher predicted probabilities. The slight divergence between women with and without experience of violence at each autonomy category is descriptive and modest, and formal interaction tests indicate that the moderating effect of violence is not statistically significant.

## Discussion

The study examined unintended pregnancy and birth (UPB) in India through a relational lens, incorporating the characteristics of couple dynamics along with socio-demographic and economic characteristics. The findings of this study demonstrate linkages between relational dynamics and UPB in India. Consistent with our theoretical framework, we found that wives' health autonomy is associated with a lower likelihood of UPB, suggesting that agency allows wives to make informed choices better aligned with their fertility intentions [34,35]. However, this autonomy does not exist in a vacuum; the mediation analysis reveals that husbands' regressive sexual attitudes significantly undermine it. This suggests that patriarchal norms act as a "gatekeeper" to reproductive rights, where the husband's attitudes are indirectly linked to higher UPB risk by shrinking the wives' negotiated space [6,36]. Autonomy may not always serve as a protective factor when regressive male attitudinal norms prevail and dominate family planning decisions [16]. These relational patterns coexist with critical socio-demographic factors that independently predict UPB, most notably the wife's age, her number of living

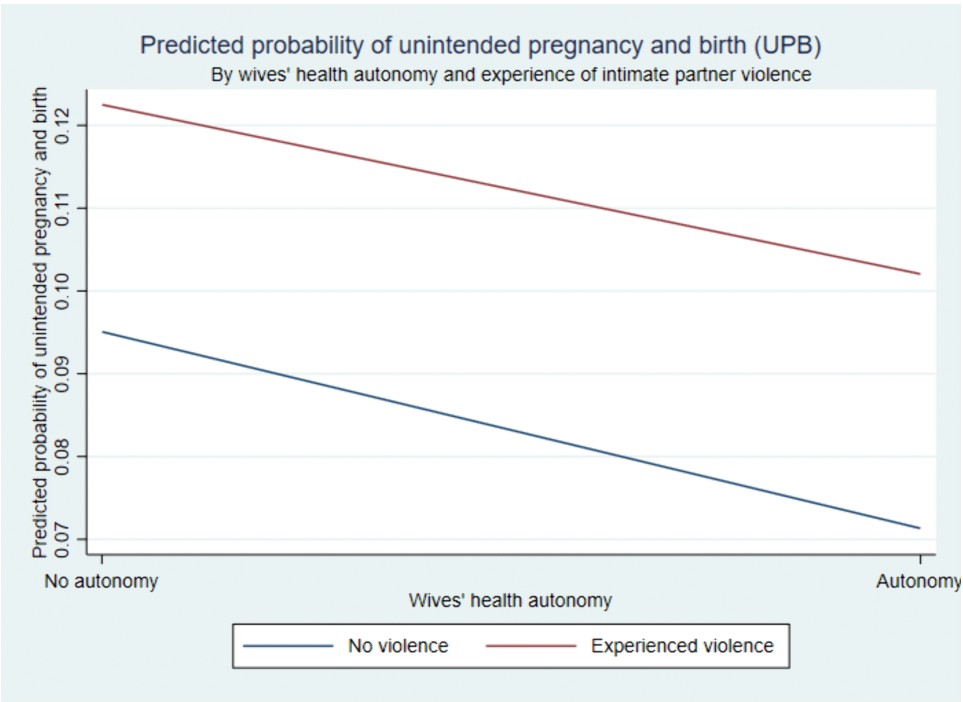

**Fig 2. Predicted probabilities of UPB by wives' health autonomy, stratified by experience of intimate partner violence.**

children, and household wealth. Our analysis showed powerful linkages between UPB and an increasing number of living children. Specifically, wives with three or more living children face more than five times the odds of UPB compared to those with no children. This pattern suggests that once a woman achieves her fertility goals, subsequent pregnancies and births are more likely to be identified as a failure of reproductive control [37,38]. Similarly, wives from the richest households are significantly less likely to experience UPB compared to those in the poorest households. This wealth gradient likely reflects better access to a wider range of modern contraceptive methods and higher consistency in their use among wealthier couples [32,39]. Wives in the 25–34-year age group experience fewer UPBs than those under 25. While this is inconsistent with prior research focused on unwanted pregnancies, which showed higher unintendedness in this middle-aged bracket [31], it aligns with recent research showing that unwanted pregnancies are lower in this age group [32]. This reflects a shift in India's family-building patterns where this age group is increasingly regarded as the most desirable stage for planned childbearing and rearing. Conversely, younger wives remain more vulnerable, which may be associated with lower contraceptive awareness and less established agency. Regarding employment, while wives working in salaried roles or agriculture have a lower prevalence of UPB, the association for salaried work was not statistically proven in the adjusted model. Notably, the significant association between agricultural work and fewer UPBs reflects context-specific mechanisms or differences in fertility goals that demand further investigation.

Intimate Partner Violence (IPV) may act as a structural constraint to negotiation. Our results show that IPV is independently associated with an increased likelihood of UPB. Such coercion not only elevates the risk of unintended conceptions [25] but also exposes the interconnection between IPV and reproductive coercion, two distinct yet closely related dynamics that undermine reproductive autonomy [40]. Even though the interaction between autonomy and violence was not statistically significant in our analysis, the persistent risk gap highlights that IPV functions as a "structural floor" of risk. These patterns are embedded in broader structures of gender inequality, where violence reinforces women's subordinate

status and the denial of reproductive rights, contributing to poor health outcomes [41]. In violent relationships, a wife's ability to act on her choices is restricted because abusive partners often exert power through reproductive control [26,42], potentially making individual empowerment efforts less effective if the violent environment is not addressed [21].

The role of education further clarifies these dynamics. In the full model adjusted for relational dynamics, wealth, and the level of safety (IPV), wives with higher levels of schooling face increased odds of UPB. In high-patriarchy contexts, education may not independently regulate reproductive outcomes if relational dynamics remain regressive [43,44]. Rather, this finding suggests that education leads to a "heightened awareness" or a change in internal standards. Educated wives are likely more sensitive to their own fertility goals and, therefore, more likely to identify and report pregnancies and births as unintended when they conflict with their defined intentions. A similar argument was established for currently pregnant women, where education was linked to higher reported unintended pregnancy [31], suggesting that education influences the subjective threshold at which a woman perceives a lack of reproductive control as a failure of her intentions. Taken together, these findings underscore that UPB is a relational outcome shaped by the interplay of power, safety, and negotiation. This aligns with the perspective that household decision-making is a complex process, where individual agency is often constrained by a partner's influence or mismatched perceptions [45]. Policy interventions must therefore move beyond the "wife-only" approach and address the embedded social structures and male attitudes that govern the household. Addressing UPB in India requires a holistic strategy that acknowledges health as an outcome of both individual agency and a safe, equitable relational environment.

## Strengths and limitations

This study has several notable strengths. First, it utilized a large, nationally representative dataset (NFHS-5), ensuring that the findings have high external validity. Second, by incorporating both wives' and husbands' characteristics within a theoretically grounded framework, the study moves beyond traditional individual-centric analyses to provide a more holistic understanding of unintended pregnancy and birth (UPB) as influenced by relational dynamics. Furthermore, the use of a multivariable adjusted model ensures the robustness of our findings and allows for a nuanced exploration of the variables involved.

However, certain limitations must be acknowledged. First, the study combined two distinct indicators: unintended pregnancy (a process currently in progress) and unintended birth (a completed outcome). The latter may be affected by post-factum rationalization, where mothers may retrospectively report a previously unintended pregnancy as intended once the child is born, potentially leading to an underestimation of UPB. Second, the NFHS-5 utilizes a cross-sectional design; therefore, our findings are conceptually based, and temporality cannot be strictly established. Third, the sensitive nature of variables such as Husband's Sexual Attitude or Intimate Partner Violence (IPV) may be influenced by social desirability bias. Finally, as our analysis is limited to currently married couples, the findings may not be generalizable to unintended pregnancies or births occurring outside of formal marital unions.

## Conclusion

This study shows that unintended pregnancy and birth in India are closely linked to couple-level dynamics, particularly women's health autonomy, husbands' sexual attitudes, and intimate partner violence. Although greater autonomy is associated with a lower likelihood of UPB, this association is weaker in relationships characterized by regressive male norms. IPV independently elevates such reproductive risk. The findings point to the limits of family planning approaches and highlight the importance of interventions that address gendered power relations in the household environment in which reproductive decisions are made and acted upon.

## Supporting information

**S1 Table. Pattern of unintended pregnancy and unintended birth across background characteristics.**
(DOCX)

   

## Acknowledgments

The authors acknowledge reviewers for their valuable insights.

## Author contributions

**Conceptualization:** Monika Lakhotia, Bhaswati Das.

**Data curation:** Monika Lakhotia.

**Formal analysis:** Monika Lakhotia.

**Methodology:** Monika Lakhotia.

**Supervision:** Bhaswati Das.

**Writing – original draft:** Monika Lakhotia.

**Writing – review & editing:** Bhaswati Das.

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
