## [Decision Letter · Decision Letter 0]

12 Jan 2026

Dear Dr. LAKHOTIA,

Thank you for submitting your manuscript to PLOS ONE. After careful consideration, we feel that it has merit but does not fully meet PLOS ONE’s publication criteria as it currently stands. Therefore, we invite you to submit a revised version of the manuscript that addresses the points raised during the review process.

**ACADEMIC EDITOR:**

Although author has tried to examine the pathways on fertility intention and behaviour in Indian context, there is much space for improvement and knowledge addition. Some of those may be bulleted as-

The dependent variable “Unintended pregnancy” is having methodological issue in defining the term. As the author has combined current pregnancy, last birth and abortion within one process variable but the outcome are different.  Author is suggested to read the extensive work by researchers from Guttmacher Institute on these three- current pregnancy, last birth and abortion related conceptual and methodological debate.
https://iussp.org/sites/default/files/Summary%20Report%20of%202021%20IUSSP%20Workshop%20on%20Measurement%20of%20Pregnancy%20Intention.pdf
https://www.guttmacher.org/journals/psrh/2003/03/measurement-and-meaning-unintended-pregnancy
The current pregnancy is a process variable but another two are the outcomes. Combining those three will have repercussion on the findings as well as suggestive measures pertaining to inform the policies.Reporting an outcome- birth as intended and/or intended are many times influenced by psychological, social and cultural context may alter the last birth either to intended or unintended. The author may go in details on it. https://pmc.ncbi.nlm.nih.gov/articles/PMC4190084/Further, what about the still birth and early neonatal death which would have been a factor of unintended process of conception, not captured by the survey data. Even including abortion aspect due to economic constrains and the last child was too young as factor of unintended pregnancy may lead to biased result pertaining to the theoretical framework author have used to examine the pathways.The theoretical framework of mediation-moderation pathways needs appropriate inclusion/exclusion of variables to measure the outcome robustly.   Further, considering Kabeer framework on intra-household power relation where agency play major role may enrich conceptualization of the study. Other than health autonomy, women attitude on gender roles may be one step ahead in shaping women autonomy and empowerment dimension in reporting any process and/or outcome variable intendedness.So, author is suggested to the revise paper on following four aspects-

Operational Definition of unintended pregnancyTheoretical framework and conceptualization on pathwaysInclusion/exclusion variables in different indices used in the studyThe model used to measure the risk of process/outcome. Author may consider separate model on current pregnancy intendedness, last birth intendedness and abortion related reasons depicting unintendedness. The final model may be developed based on the 3 components if considered as unintended pregnancy

Within the predictors, considering spousal educational difference may be more influential in the framework author used in the study.Further, measuring mediation pathways considering gender roles within the autonomy framework seemingly have somewhat different estimates.Based on the refined analysis, the result and findings will be somewhat different. So, author is advised to interpret with caution. Accordingly the mediation-moderation pathways need appropriate interpretation and conclusion.Thorough editing is also expected in the revised version of the manuscript.

We look forward to receiving your revised manuscript.

Kind regards,

Rajesh Raushan, PhD

Academic Editor

PLOS One

Journal Requirements:

3. We note that Figure 2 in your submission contain copyrighted images. All PLOS content is published under the Creative Commons Attribution License (CC BY 4.0), which means that the manuscript, images, and Supporting Information files will be freely available online, and any third party is permitted to access, download, copy, distribute, and use these materials in any way, even commercially, with proper attribution. For more information, see our copyright guidelines: http://journals.plos.org/plosone/s/licenses-and-copyright.

Additional Editor Comments :

None

Reviewers' comments:

Reviewer's Responses to Questions

**Comments to the Author**

1. Is the manuscript technically sound, and do the data support the conclusions?

Reviewer #1: Yes

Reviewer #2: Yes

2. Has the statistical analysis been performed appropriately and rigorously?

Reviewer #1: Yes

Reviewer #2: Yes

3. Have the authors made all data underlying the findings in their manuscript fully available?

Reviewer #1: Yes

Reviewer #2: Yes

4. Is the manuscript presented in an intelligible fashion and written in standard English?

Reviewer #1: Yes

Reviewer #2: Yes

Reviewer #1: The manuscript addresses an important and policy-relevant question. It examines how couple power dynamics on women’s health autonomy and husbands’ sexual attitudes shape unintended pregnancies in India using NFHS-5 couple data, applying appropriate methods. In particular, this paper focuses explicitly on mediation and moderation pathways, moving beyond simple association models and highlighting indirect effects of male attitudes via women’s health autonomy. The authors may consider the following comments to be incorporated into their manuscript.

• The theoretical section is rich but somewhat dense, and the explicit hypotheses are not clearly stated. Please add a short subsection at the end of the theoretical framework that explicitly lists the main hypotheses (e.g., H1–H4 on autonomy, attitudes, IPV, mediation, and moderation) to guide the reader and align with the analytic strategy.

• The outcome variable “unintended pregnancy” combines current pregnancies, recent births, and abortions due to several reasons (e.g., contraceptive failure, economic constraints). Reporting “wanted later” and “wanted no more” for delivered births may be affected by post-factum rationalisation. Therefore, the authors may consider including a sensitivity analysis based on the difference between the desired and actual number of children. For theoretical and practical reference, the authors may consult “Rana, M. J., Cleland, J., Sekher, T. V., & Padmadas, S. S. (2021). Disentangling the effects of reproductive behaviours and fertility preferences on child growth in India. Population Studies, 75(1), 37–50”.

• In Table 3 and Figure 1, the direction of the associations between variables is not explicitly indicated. The authors may consider revising these to make the tables and figures more self-explanatory.

• The authors may provide more detailed justification for the categorisation of the independent variables capturing couple dynamics.

• The cross-sectional nature of NFHS-5 limits causal inference, yet some parts of the Discussion use language that may be interpreted as causal. The authors may add a brief paragraph acknowledging that mediation and moderation are modelled conceptually and that temporality cannot be established due to the cross-sectional design.

• In Table 3, p-values should not be reported as 0; these should be replaced with “<0.001” where appropriate.

• The Introduction and Discussion could be slightly streamlined to reduce repetition and to break very long paragraphs into shorter, more focused ones.

Reviewer #2: The paper is well researched and addresses a highly relevant topic under the fertility section, which is one of the most important domains of demography. Unwanted pregnancy often occurs due to unmet need for contraception or inadequate knowledge regarding conception and pregnancy. Couple dynamics are closely associated with unwanted pregnancy, and this relationship is well reflected in the paper. The methodology is systematic, and the approach adopted to ensure privacy and confidentiality during analysis is highly commendable. The findings of this study will be useful for addressing the identified barriers contributing to unwanted pregnancy.

However, the following concerns need to be addressed:

1. In the methodology section under Data Source, revise the sentence to state that NFHS-5 is a large-scale survey.

2. Maintain consistency by reporting numerical values up to one decimal place throughout the manuscript, including all tables.

3. In Table 1, the percentage of unwanted pregnancy is reported as weighted; however, it is not clear whether the total number of pregnancies (sample size) is weighted or unweighted. Please clarify this in the table or its footnote.

4. In Table 3, reporting both p-values and significance stars (***) is redundant. Please retain either the p-value or the significance stars, and ensure consistency in both the table and the text. If p-values are retained, report them up to three decimal places throughout the manuscript.

5. Although limitations are discussed within the manuscript, please present Limitations and Strengths as a separate, clearly defined section.

6. Present the Conclusion as a separate section with an appropriate heading.

7. Include Policy Implications and Future Research Directions based on the key findings of the study, if applicable.

**Do you want your identity to be public for this peer review?** For information about this choice, including consent withdrawal, please see our Privacy Policy

Reviewer #1:**Yes**

Reviewer #2:**Yes**

---

## [Author Response · Author response to Decision Letter 1]

29 Jan 2026

Response to Editor and Reviewers-

We sincerely thank the Editor and the reviewers for their careful reading of the manuscript and for their constructive and insightful comments. The suggestions have substantially improved the clarity, rigour, and overall quality of the paper. All comments have been carefully considered, and the manuscript has been revised accordingly. Our point-by-point responses are provided below.

Journal Requirements:

Response from Author – All necessary changes have been made to ensure that the manuscript complies with PLOS ONE’s style requirements, including file naming.

Response from Author – The source of the data has now been clearly specified in the Data Availability Statement.

3. We note that Figure 2 in your submission contains copyrighted images. All PLOS content is published under the Creative Commons Attribution License (CC BY 4.0), which means that the manuscript, images, and Supporting Information files will be freely available online, and any third party is permitted to access, download, copy, distribute, and use these materials in any way, even commercially, with proper attribution.

Response from Author – Figure 2 was prepared by the author using analysed data and does not include copyrighted material. As the analysis has been revised, the earlier figure has been replaced with a newly prepared figure, which is also entirely created by the author.

Comments of Editor

The author is suggested to revise the paper on the following four aspects-

1. Operational Definition of Unintended Pregnancy-

Response from Author – The outcome variable has been revised and is now termed Unintended Pregnancy and Birth (UPB). The revised manuscript clearly explains the construction of this variable.

2. Theoretical framework and conceptualization on pathways

Author’s Response - The theoretical framework has been restructured by synthesizing relevant theories more explicitly, and the conceptual framework has been revised accordingly to clarify the proposed pathways.

3. Inclusion/exclusion variables in different indices used in the study

Response from Author – The construction of all indices has now been described in detail. As the component, abortion has not been used- the variable describing contraceptive decision making is removed in the revised analysis, and unwanted births are retrospectively reported.

4. The model used to measure the risk of process/outcome. The author may consider a separate model on current pregnancy intendedness, last birth intendedness and abortion related reasons depicting unintendedness. The final model may be developed based on the 3 components if considered as an unintended pregnancy

Response from Author – Following the Editor’s suggestion, the outcome variable has been revised to Unintended Pregnancy and Birth (UPB), combining current unintended pregnancies and unintended last births within five years preceding NFHS-5. Abortion-related unintendedness has been excluded.

Because the analysis is based on the couple file, the sample size for the current pregnancy alone was small, leading to sparse cell issues. To ensure robustness, we conducted separate bivariate analyses for the two components- current unintended pregnancy and unintended last birth, using the Rao–Scott chi-square test. These results are provided as Supplementary Information (S1 Table). The limitation related to post-factum rationalisation of unintended births has been explicitly acknowledged in the revised manuscript.

As the analysis has been substantially revised, the manuscript has undergone thorough editing.

Comments of Reviewer 1:

1. The theoretical section is rich but somewhat dense, and the explicit hypotheses are not clearly stated. Please add a short subsection at the end of the theoretical framework that explicitly lists the main hypotheses (e.g., H1–H4 on autonomy, attitudes, IPV, mediation, and moderation) to guide the reader and align with the analytic strategy.

Response from Author- At the end of the theoretical framework, all the hypotheses are added.

2. The outcome variable “unintended pregnancy” combines current pregnancies, recent births, and abortions due to several reasons (e.g., contraceptive failure, economic constraints). Reporting “wanted later” and “wanted no more” for delivered births may be affected by post-factum rationalisation.

Response from Author – The outcome variable has been revised to Unintended Pregnancy and Birth (UPB), abortion-related components have been removed, and the limitation of post-factum rationalisation has been explicitly discussed in the Limitations section.

3. In Table 3 and Figure 1, the direction of the associations between variables is not explicitly indicated. The authors may consider revising these to make the tables and figures more self-explanatory.

Response from Author – We used arrows to show the pathway linkages, and the association is described through beta coefficients (β )

4. The authors may provide more detailed justification for the categorisation of the independent variables capturing couple dynamics.

Response from Author – Incorporated into revised manuscript.

5.The cross-sectional nature of NFHS-5 limits causal inference, yet some parts of the Discussion use language that may be interpreted as causal. The authors may add a brief paragraph acknowledging that mediation and moderation are modelled conceptually and that temporality cannot be established due to the cross-sectional design.

Response from Author – The limitations of the cross-sectional design have been clearly acknowledged, and the Discussion has been revised to avoid causal interpretations. Mediation and moderation are now explicitly framed as conceptual and associative.

6. In Table 3, p-values should not be reported as 0; these should be replaced with “<0.001” where appropriate.

Response from Author – P-values are no longer reported as zero. Instead, levels of statistical significance are indicated using *** p<0.01, ** p<0.05, and * p<0.10.

7. The Introduction and Discussion could be slightly streamlined to reduce repetition and to break very long paragraphs into shorter, more focused ones.

Response from Author – The Introduction and Discussion have been streamlined, with long paragraphs broken into shorter, more focused sections to reduce repetition.

Comments of Reviewer 2:

1. In the methodology section under Data Source, revise the sentence to state that NFHS-5 is a large-scale survey.

Response from Author – It is mentioned in the data source (line number- 153)

2. Maintain consistency by reporting numerical values up to one decimal place throughout the manuscript, including all tables.

Response from Author – Numerical values are consistently reported up to two decimal places throughout the manuscript, including all tables, to improve precision of confidence intervals.

3. In Table 1, the percentage of unwanted pregnancy is reported as weighted; however, it is not clear whether the total number of pregnancies (sample size) is weighted or unweighted. Please clarify this in the table or its footnote.

Response from Author – It is now clearly stated in Table 1 that percentages are weighted, while the sample size is unweighted.

4. In Table 3, reporting both p-values and significance stars (***) is redundant. Please retain either the p-value or the significance stars, and ensure consistency in both the table and the text. If p-values are retained, report them up to three decimal places throughout the manuscript.

Response from Author – P-values have been removed, and statistical significance is consistently reported using significance stars only.

5. Although limitations are discussed within the manuscript, please present Limitations and Strengths as a separate, clearly defined section.

Response from Author – The section is added.

6. Present the Conclusion as a separate section with an appropriate heading.

Response from Author – A separate section for the conclusion has been added.

7. Include Policy Implications and Future Research Directions based on the key findings of the study, if applicable.

Response from Author – We thank the reviewer for this suggestion. Rather than adding separate sections, policy implications have been integrated within the Discussion and Conclusion, where they are directly linked to the study’s key findings.

---

## [Decision Letter · Decision Letter 1]

20 Feb 2026

Beyond individual choice : Exploring pathways linking couple dynamics with unintended pregnancy and birth in India

PONE-D-25-63031R1

Dear Ms. LAKHOTIA,

We’re pleased to inform you that your manuscript has been judged scientifically suitable for publication and will be formally accepted for publication once it meets all outstanding technical requirements.

Kind regards,

Rajesh Raushan, PhD

Academic Editor

PLOS One

Additional Editor Comments (optional):

Reviewers' comments:

Reviewer's Responses to Questions

**Comments to the Author**

Reviewer #2: All comments have been addressed

2. Is the manuscript technically sound, and do the data support the conclusions?

Reviewer #2: Yes

3. Has the statistical analysis been performed appropriately and rigorously?

Reviewer #2: Yes

4. Have the authors made all data underlying the findings in their manuscript fully available?

Reviewer #2: Yes

5. Is the manuscript presented in an intelligible fashion and written in standard English?

Reviewer #2: Yes

Reviewer #2: All reviewer comments were addressed by authors is satisfactorily; the manuscript is recommended for publication

**Do you want your identity to be public for this peer review?** For information about this choice, including consent withdrawal, please see our Privacy Policy

Reviewer #2: **Yes:** Dr Jeetendra Yadav

---

## [Editor Report · Acceptance letter]

PONE-D-25-63031R1

PLOS One

Dear Dr. LAKHOTIA,

I'm pleased to inform you that your manuscript has been deemed suitable for publication in PLOS One. Congratulations! Your manuscript is now being handed over to our production team.

Kind regards,

on behalf of

Dr. Rajesh Raushan

Academic Editor

PLOS One